# Dysbiosis in Human Urinary Microbiota May Differentiate Patients with a Bladder Cancer

**DOI:** 10.3390/ijms251810159

**Published:** 2024-09-21

**Authors:** Julie A. Vendrell, Simon Cabello-Aguilar, Romain Senal, Elise Heckendorn, Steven Henry, Sylvain Godreuil, Jérôme Solassol

**Affiliations:** 1Laboratoire de Biologie des Tumeurs Solides, Département de Pathologie et Oncobiologie, CHU Montpellier, Université de Montpellier, 34295 Montpellier, France; j-vendrell@chu-montpellier.fr (J.A.V.); s-cabelloaguilar@chu-montpellier.fr (S.C.-A.); r-senal@chu-montpellier.fr (R.S.);; 2Montpellier BioInformatique pour le Diagnostic Clinique (MoBiDiC), Plateau de Médecine Moléculaire et Génomique (PMMG), CHU Montpellier, 34295 Montpellier, France; 3Institut du Cancer de Montpellier (ICM), Département de Biopathologie, 34295 Montpellier, France; 4Laboratoire de Bactériologie, CHU Montpellier, Université de Montpellier, 34295 Montpellier, France; steven-henry@chu-montpellier.fr (S.H.); s-godreuil@chu-montpellier.fr (S.G.); 5Institut Régional du Cancer de Montpellier (IRCM), Université de Montpellier, ICM, Inserm, 34298 Montpellier, France

**Keywords:** bladder cancer, urinary microbiota, NGS, *Rhodanobacter*, *Cutibacterium*

## Abstract

Recent interest in noninvasive diagnostic approaches has highlighted the potential of urinary microbiota as a novel biomarker for bladder cancer. This study investigated the urinary microbiota of 30 bladder cancer patients and 32 healthy controls using a specific NGS protocol that sequences eight hypervariable regions of the 16S rRNA gene, providing detailed insights into urinary microbiota composition. The relative abundance of microbial compositions in urine samples from cancer patients and healthy controls was analyzed across various taxonomic levels. No notable differences were highlighted at the phylum, class, order, and family levels. At the genus level, 53% of detected genera were represented in either cancer patients or healthy controls. Microbial diversity was significantly lower in cancer patients. The differential analysis identified five genera, *Rhodanobacter*, *Cutibacterium*, *Alloscardovia*, *Moryella*, and *Anaeroglobus*, that were significantly more abundant in cancer patients. Notably, *Rhodanobacter* was present in 20 cancer samples but absent in healthy controls. Conversely, 40 genera, including Lactobacillus, *Propionibacterium*, and *Bifidobacterium*, exhibited reduced abundance in cancer patients. These findings suggest that some genera may serve as potential biomarkers for bladder cancer, highlighting the need for further research to explore their roles in disease pathogenesis and their potential applications in diagnostics and therapeutics.

## 1. Introduction

Bladder cancer affects over 600,000 individuals annually, resulting in more than 220,000 deaths worldwide, making it the ninth most prevalent malignancy [1]. Its incidence escalates significantly with age, predominantly affecting those over 65 years old. Given the aging global population, the incidence of bladder cancer is anticipated to rise substantially in the coming years. Demographically, bladder cancer is diagnosed 3.7 times more frequently in men than in women [2]. Although the exact etiology remains incompletely understood, genetic mutations and environmental factors such as tobacco smoking, exposure to carcinogens, chlorinated drinking water, and cyclophosphamide are recognized as major risk factors in bladder cancer development [3]. Traditional diagnostic methods, including cystoscopy and imaging, present certain limitations, prompting the search for noninvasive alternatives.

Over the last decade, the scientific community has increasingly acknowledged the pivotal role of the human microbiome in maintaining health and contributing to disease pathogenesis. High-throughput DNA sequencing technologies, particularly the democratization of next-generation sequencing (NGS), have revolutionized microbiome research, unveiling complex associations between microbial communities and a variety of diseases. Although the majority of studies have focused on the gut microbiota, emerging evidence suggests that the urinary tract microbiome also plays a crucial role in health and disease, including cancer [4].

The association between microorganisms and cancer development is well-documented, with estimates suggesting that pathogens may contribute to approximately 20% of human malignancies [5]. The interaction between microorganisms and their human hosts is highly intricate, influencing cancer development, progression, and response to treatment through various molecular mechanisms [6]. Certain bacteria can directly induce host DNA damage through the production of genotoxins or the generation of reactive oxygen species [7]. Additionally, pathogens can modulate host signaling pathways, such as the Wnt/β-catenin or MAPK pathways, thereby contributing to cancer development and progression [8]. Moreover, the microbiome can induce chronic inflammation, creating a microenvironment conducive to tumor development, or trigger immunosuppressive responses that undermine cancer immune surveillance [9,10].

Focusing more precisely on the urinary microbiota, dysbiosis has been implicated in various urological disorders, including infections [11], urinary incontinence [12,13], kidney transplant rejection, or dysfunction [12,14], and has been associated with cancer [15,16,17]. Additionally, therapeutic interventions such as the intravesical instillation of *Mycobacterium bovis* bacillus Calmette–Guérin (BCG) and the oral administration of *Lactobacillus* post-tumor removal have been shown to reduce recurrence rates [18,19]. Dysbiosis resulting from repeated antibiotic use has also been linked to the increased incidence of cancers, including bladder cancer [20]. These findings raise the possibility that alterations in the urinary microbiota may also be associated with bladder cancer initiation and development.

Collectively, these observations support the hypothesis that the microbiome may influence bladder carcinogenesis, progression, and relapse. Despite extensive research into the impact of the human gut microbiome on health and disease, including various cancers [21], comprehensive analyses of the urinary microbiota of bladder cancer remain underreported. Recent studies have indicated that the urinary microbiota of bladder cancer patients may be distinct from that of healthy volunteers [15,16,17,22,23]. However, discrepancies in these results likely arise from the use of non-standardized protocols for studying urinary microbiota and the lack of consensus in NGS and bioinformatics analysis methods.

In our study, we used a specific NGS protocol previously established by our team [24] that enables the sequencing of eight out of nine hypervariable regions of the 16S rRNA gene, providing more comprehensive information than previous studies. This methodology was applied to accurately identify bacteria abundance in urine samples from 30 patients with bladder cancer and 32 healthy volunteers. The bacterial taxa identified in our study may be implicated in the pathogenesis of bladder cancer, and further exploration could lead to the development of novel diagnosis tools or therapeutic approaches.

## 2. Results

### 2.1. Sequencing Analysis of Human Urine Microbiota

Following NGS and bioinformatics analysis, samples with less than 2000 operational taxonomic units (OTUs) at the genus level were removed from further analysis, resulting after the exclusion of non-informative samples in a final cohort comprising 30 urine samples from patients with bladder cancer and 32 specimens from healthy volunteers. The cases and controls were matched by age and sex (Table 1). Among the bladder cancer patients, 27 had a non-muscle invasive bladder cancer (NMIBC) and 18 had received Bacillus Calmette Guerin (BCG) therapy.

To minimize potential bias in the comparative analysis of samples, particularly those with a high number of reads, we conducted 10 random reads subsampling per sample, limiting the total number of OTUs to 2000. For each sample, the mean value of the 10 resulting matrices was calculated and used for the subsequent analyses (Appendix A). The resulting OTUs were then analyzed across multiple taxonomic levels, including phylum, class, order, family, and genus. Furthermore, to enhance the robustness of the analysis, taxa with fewer than 10 OTUs across all samples were excluded from the dataset (Appendix A).

### 2.2. Relative Abundance of Urine Bacteria at Different Taxa Levels

The relative abundance of microbial taxa in urine samples from both cancer patients and healthy volunteers was displayed at the phylum, class, order, family, and genus level (Figure 1). At the phylum level, the most prevalent taxa were *Proteobacteria*, *Firmicutes*, and *Actinobacteria*. At the class level, *Bacilli*, *Actinobacteria*, *Clostridia*, and *Gammaproteobacteria* were the most dominant. At the order level, *Lactobacillales*, *Peptostreptococcales-Tissierellales*, and *Corynebacteriales* were prominent, while at the family level, *Peptostreptococcales-Tissierellales_fa*, *Corynebacteriaceae*, and *Streptococcaceae* were the most represented (Appendix A). At the genus level, more pronounced differences in the relative abundance of taxa were observed between cancer and control samples (Figure 1E), as well as between individual samples within each group (Figure 2A). Although the urinary microbiota in both groups was predominantly composed of *Corynebacterium* and *Streptococcus*, it is noteworthy that 53% of the detected genera are represented at a relative abundance of more than 1% in only one group, either in the cancer patients or the healthy volunteers (Appendix A). Furthermore, a statistically significant difference in alpha diversity was observed at the genus level, with higher microbial diversity observed in control samples compared to cancer samples (Shannon index, Mann–Whitney test, *p* < 0.0005, Figure 2B).

### 2.3. Differential Urinary Microbiota Abundance Profiles between Bladder Cancer Patients and Healthy Volunteers

The differential analysis of taxonomic abundance between urine samples from bladder cancer patients and healthy volunteers was conducted using DESeq2, a statistical method that estimates variance-mean dependence in count OTUs, modeled through a negative binomial distribution. Although the overall relative abundance of most species was similar between the two groups, several significant differences in microbiota composition were observed (Figure 3 and Appendix A). Specifically, 5 genera out of the 186 detected (2.7%) were significantly more abundant in urine samples from cancer patients. These genera, ranked from most to least significant, include *Rhodanobacter*, *Cutibacterium*, *Alloscardovia*, *Moryella*, and *Anaeroglobus* (Figure 3). Of most interest, *Rhodanobacter* was detected in 20 urine samples from cancer patients but was completely absent in samples from healthy volunteers, while *Cutibacterium* was present in 18 samples from cancer patients and only 3 samples from healthy volunteers, highlighting a significant disparity in the presence of these bacteria between the two groups of samples (χ^2^ test, *p* < 1.10^−7^ and *p* = 0.007, respectively, Appendix A). Conversely, 40 genera were found to be significantly less abundant in urine samples from cancer patients, representing 21.5% of the detected genera. The most significantly underrepresented genera in cancer patients included *Lactobacillus*, *Propionibacterium*, *Porphyromonas*, *Atopobium*, *Prevotella*, and *Bifidobacterium* (Figure 3).

## 3. Discussion

The urinary microbiota has been under intense scrutiny in recent years, with accumulating evidence linking it to the development of specific cancers, including bladder cancer. While several studies have investigated the relationship between urinary microbiota and bladder cancer, there is currently limited information regarding the complex interplay between bladder cancer and the urinary microbiome. In our study, we conducted a comprehensive analysis of urine samples from 30 patients who had developed bladder cancer and 32 healthy volunteers. To obtain more detailed insights than previous studies, we employed a specific NGS protocol previously validated by our group [24], which allows the sequencing of eight out of the nine hypervariable regions of the 16S rRNA gene. This advanced approach resulted in a significant reduction in non-attributed reads compared to the findings reported in other studies [25,26], thereby enhancing the detection of specific genera. Using this methodology, we observed a markedly lower diversity in bladder cancer (Shannon index). Currently, the alpha-diversities reported in other studies concerning the urine microbiota remain controversial. For instance, Chipollini et al. [27] reported lower diversity in bladder cancer samples, whereas Bi et al. [15] observed an increase in diversity and Wu et al. [17] and Hussein et al. [22] did not observe any significant difference in alpha-diversity. Notably, a study conducted on bladder tissue also identified a significant decrease in diversity within cancerous tissue compared to normal bladder tissue [28]. These discrepancies may be attributed to the use of non-standardized protocols for urinary microbiota analysis and the lack of consensus regarding bioinformatics methodologies.

A more thorough examination of the genera significantly enriched in urine from control patients revealed a predominance of commensal genera. In alignment with the findings of previous studies [15,28], our work identified two well-documented probiotics, *Lactobacillus* and *Bifidobacterium*, as some of the most significantly overrepresented. Notably, three randomized clinical trials have demonstrated that *Lactobacillus casei* probiotic supplementation reduces the recurrence of bladder cancer in patients [18,29,30]. Furthermore, the administration of *Lactobacillus* and *Bifidobacterium* in bladder cancer-xenografted mice enhanced the efficacy of chemotherapy [31]. A randomized clinical trial is currently underway to evaluate the efficacy of a probiotic mixture composed of several *Lactobacillus*, *Bifidobacterium*, and *Enterococcus* strains on immunotherapy of urothelial carcinoma (NCT05220124). These findings suggest a potential protective role of these bacteria against bladder cancer and indicate that probiotics could serve as a promising therapeutic strategy for bladder cancer patients.

Our analyses also revealed a significant enrichment of *Rhodanobacter*, *Cutibacterium*, *Alloscardovia*, *Moryella*, and *Anaeroglobus* in urine samples from bladder cancer patients. Importantly, previous studies have established associations between the presence of these bacteria and various cancers. For instance, *Rhodanobacter* has been found to be more abundant in tissue biopsies from colorectal cancer patients than in those with adenomatous polyps and in those with adenomatous polyps compared to healthy controls [32]. Additionally, gastric microbiota analysis revealed an increased prevalence of *Rhodanobacter* in gastric cancer patients compared to individuals with superficial gastritis [33]. *Cutibacterium* has been identified as prevalent and abundant in bladder cancer tissue [34,35] and its abundance was correlated with the effectiveness of BCG treatment [36]. Furthermore, *Cutibacterium acnes* was detected in extracellular vesicles from patients with renal cell carcinoma, and these vesicles were shown to promote tumorigenesis in a mouse model of renal cancer allografts [37]. In our study, for the first time to our knowledge, we report that the detection of *Rhodanobacter* and *Cutibacterium* in urine samples might serve as highly specific biomarkers for bladder cancer. Further explorations are warranted to unravel the potential role of these bacteria in bladder tumorigenesis.

Regarding the other genera that were significantly more abundant in urine samples from cancer patients, *Alloscardovia* was enriched in the gut microbiota of patients with intrahepatic cholangiocarcinoma compared to those in patients with hepatocellular carcinoma, liver cirrhosis, and healthy individuals [38]; *Moryella* was more abundant in the mucosal tissue of patients with early gastric neoplasia compared to those with gastric intestinal metaplasia [39]; and *Anaeroglobus* was significantly enriched in the saliva of patients with thyroid cancer compared to healthy volunteers [40].

In conclusion, our findings indicate that bacteria overrepresented in the urine of patients with bladder cancer, such as *Rhodanobacter*, *Cutibacterium*, *Alloscardovia*, *Moryella*, and *Anaeroglobus*, may serve as novel potential biomarkers for bladder cancer. The identification of specific bacterial signatures associated with bladder cancer could improve non-invasive diagnostic accuracy and facilitate early cancer detection. Additionally, the higher abundance of these bacteria in the urine of cancer patients may suggest a contributory role in cancer pathogenesis; investigating their potential as therapeutic targets could lead to innovative treatment strategies for bladder cancer. Comprehensive studies are required to elucidate the precise mechanisms linking these bacteria to bladder cancer. Understanding how they contribute to tumorigenesis, whether through direct interactions with host cells, modulation of the immune response, or other mechanisms, remains a significant challenge. Ultimately, pinpointing the role of specific microbes in bladder cancer pathogenesis is a complex endeavor that necessitates further investigations. Our findings lay the groundwork for future research to unravel these intricate relationships, with the ultimate aim of improving the diagnosis and treatment of bladder cancer.

## 4. Methods

### 4.1. Urine Sample Collection

Patients undergoing consultation at the hospital between January 2020 and October 2020 following a diagnosis of bladder cancer were recruited for this study. Midstream urine samples were obtained from 36 healthy volunteers and 46 bladder cancer patients. Urine from bladder cancer patients was collected before surgery or biopsy. Samples were immediately transferred to sodium borate tubes. A volume of 15 mL of urine per sample was centrifuged at 5000× *g* for 10 min at 4 °C, and the resulting pellet was stored at −20 °C until DNA extraction.

### 4.2. DNA Extraction and Quantification

DNA extraction was performed using the Qiagen DNeasy Blood and Tissue kit, according to the manufacturer’s recommendations, as previously described [24]. Additional pretreatment steps for the extraction of Gram-positive bacteria were included. DNA was eluted in 50 µL of elution buffer and quantified using a Qubit 2.0 fluorometer with the Qubit dsDNA HS Assay Kit (Thermo Fisher Scientific, Waltham, MA, USA).

### 4.3. Next-Generation Sequencing (NGS) Experiments

PCR amplification targeted the V1–V3, V3–V4, V4–V5, and V6–V8 regions of the 16S rRNA gene (Appendix A). The regions of interest were amplified using the Access Array (Fluidigm, San Francisco, CA, USA) in conjunction with a 48.48 Fluidigm Access Array System. The resulting libraries were indexed, pooled, and qualified by the use of D1000 ScreenTapes and a 4200 TapeStation instrument (Agilent Technologies, Santa Clara, CA, USA). Quantification was performed with the KAPA Library Quantification Kit (Roche, Basel, Switzerland) on a LightCycler 480 (Roche). Pair-end sequencing (2 × 300 cycles) was subsequently conducted on a MiSeq instrument (Illumina, San Diego, CA, USA).

### 4.4. Bioinformatics Analysis

Raw sequence data were processed and filtered with DADA2 (v1.16) [41] on Rstudio. Briefly, primers were cut using the *cutadapt* function [42], and reads with a quality score below 30 were truncated using the *filterAndTrim* function. The paired forward and reverse reads were then merged. Taxonomic assignment was performed by comparing sequences against the SILVA reference database using the assign taxonomy function, generating operational taxonomic units (OTUs) at the phylum, class, order, family, and genus levels [43] (Appendix A). For subsequent statistical analysis, all non-attributed OTUs at each level were removed.

### 4.5. Statistical Analysis

Statistical analyses were conducted using the R package, DESeq2 (v1.20.0) [44], with the *betaPrior* option enabled. OTUs with *p*-values ≤ 0.05 and an absolute value of log2 fold change ≥ 1 were considered significantly differentially abundant between cancer and control samples.

## Figures and Tables

**Figure 1 ijms-25-10159-f001:**
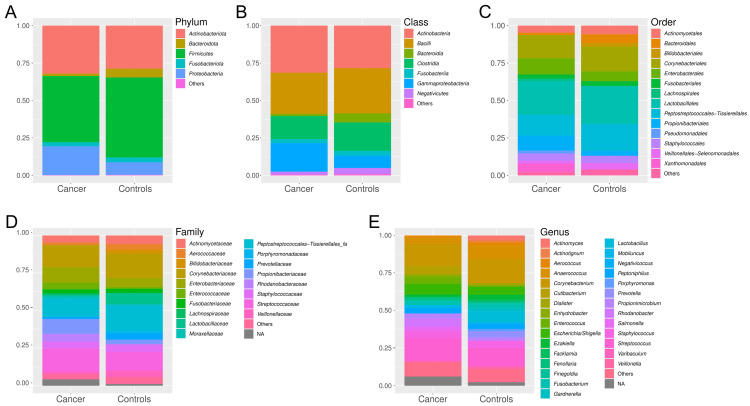
Average relative abundance of microbial taxa in urine samples from bladder cancer patients and healthy volunteers. Taxa with a relative abundance of <1% are grouped under “Others”. The percent community composition is shown at the phylum (**A**), class (**B**), order (**C**), family (**D**), and genus (**E**) levels. NA, non-attributed operational taxonomic units (OTUs).

**Figure 2 ijms-25-10159-f002:**
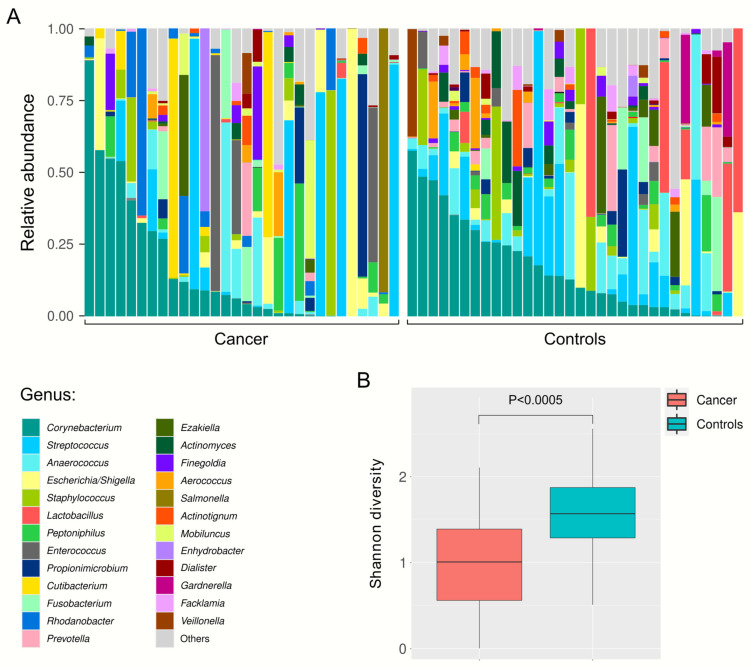
Urinary microbiota composition of samples at the genus level. (**A**) Relative abundance of microbial taxa in individual urine samples from bladder cancer patients and healthy volunteers. Taxa with a relative abundance of <1% are grouped under “Others”. (**B**) Shannon diversity index of urine samples from bladder cancer patients and healthy volunteers. Box plots display the median and range of diversity measures.

**Figure 3 ijms-25-10159-f003:**
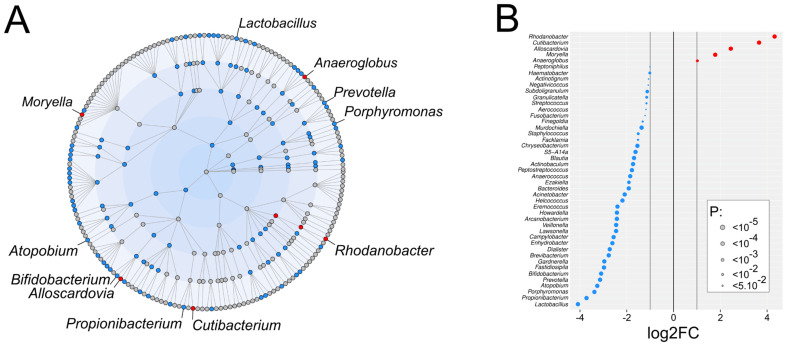
Differential urinary microbiota abundance profiles between bladder cancer patients and healthy volunteers. (**A**) Cladogram showing the differentially represented taxa across taxonomic ranks. Each circle corresponds to a taxonomic level; from the center outward: phylum, class, order, family, and genus. Blue dots indicate taxa underrepresented in the urine samples of bladder cancer patients; red dots indicate taxa that are overrepresented. Grey dots represent taxa detected in both groups without significant differences. (**B**) Dot plot illustrating the genera with significant differential representation between the two groups. Genera are considered significant if their |log2 fold change| is ≥1 and the *p*-value ≤ 0.05.

**Table 1 ijms-25-10159-t001:** Clinical and demographic characteristics of the cohort.

Clinical and Demographic Parameters		Cancer Patients (n = 30)	Healthy Volunteers (n = 32)	*p*
Age at sampling (y: mean ± SD)		72.6 ± 10.0	70.6 ± 13.8	NS (0.37) ^a^
Sex	Male	24	21	NS (0.21) ^b^
	Female	6	11	
Smoking history				
	Have smoked	4	NA	
	Smoker	11		
	Non-smoker	13		
History of BCG therapy				
	Yes	18	NA	NA
	No	10		
	Unknown	2		
Age at cancer diagnosis (y: mean ± SD)		70.4 ± 10.8	NA	NA
Invaded muscle at bladder cancer diagnosis				
	NMIBC	27	NA	
	MIBC	2		
	Unknown	1		

^a^ Student *t*-test; ^b^ χ^2^ test. *p*-value was considered as significant when *p* < 0.05. NS, not significant. NA, not applicable. NMIBC, non-muscle invasive bladder cancer. MIBC, muscle-invasive bladder cancer.

## Data Availability

Data are presented in the Appendix A and available upon request.

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
