# Peer review of "Dysbiosis in Human Urinary Microbiota May Differentiate Patients with a Bladder Cancer"

_ijms, 2024, doi:10.3390/ijms251810159_

Round 1

Reviewer 1 Report

Comments and Suggestions for Authors

Congratulations to the authors for the submission entitled: "Dysbiosis in human urinary microbiota may differentiate patients with a bladder cancer". 

We are just starting exploring this field in a clinical scenario, and find it very interesting. 

In my opinion it is well structured an clear in all the aspects, however a few things need a revision:

line 87: to accurately ?? bacteria (identify?, measure maybe?)

line 94: please define OTUs (operational taxonomic units) as it appears first time in the paper.

Foot of table 1: NS, not significant

Page 8, scheme 1. Where is the reference in the text? Is it part of Fig 3?

Lines 246-247:

When was urine collection performed in the patient group, before the operation at the office, or during intervention?. Of course after the operation it would not be appropriate.

Which population was included in healthy volunteers?

Didn't you collect any tissue sample to compare with urine?

Author Response

We would like to sincerely thank the reviewer for her/his positive comments. Additionally, we greatly appreciate the careful review of our paper, which highlighted several typographical errors. We have corrected all of them in the revised version of the manuscript. We also clarified the point regarding urine collection by adding the following sentence: “Urine from bladder cancer patients was collected before surgery or biopsy. Urinary samples from healthy volunteers with no previous history of bladder cancer and no urinary infection at the time of collection were included in the study.

Regarding Scheme 1 on page 8, it corresponds to our Supplementary Figure 1, which is cited in the text. Its title (Scheme 1) was changed by the editorial office. I will check with them to ensure consistency between the text and the figure legend. In any case, this figure will be removed in the final version of the manuscript and added to the supplementary material files.

Although analyzing the bacteria in the corresponding tissue samples would have been highly valuable, biopsy specimens were unfortunately unavailable, as they were reserved for diagnostic purposes.

Reviewer 2 Report

Comments and Suggestions for Authors

A very interesting research article that approaches a subject that starts to raise more and more interest in the research community.

The paper is titled “Dysbiosis in human urinary microbiota may differentiate patients with a bladder cancer” – a very simple but very comprehensive title, as the purpose of the paper was indeed to find if there is any change in the microbiota of bladder cancer patients which might be used as a marker for early diagnosis or if it can be changed in order to help with treatment.

Introduction was well composed and presented, and made a great preamble to the paper. It presents the significance of bladder cancer in the population, the risk factors, and also the influence of the microbiota on this pathology. The authors also present the premises for why their research is relevant. There is a small mistake on page 2 rows 86-87, the sentence “This methodology was applied to accurately bacteria abundance in urine samples from 30 patients with bladder cancer and 32 healthy volunteers “ misses a word and because of that it makes no sense. I think the authors wanted to say "to accurately identify bacteria abundance.." .

The paper starts in a more unusual way, firstly presenting the results of the research. The authors presented in a very professional matter the results of the sequencing of urinary microbiota, relative abundance of urine bacteria at different taxa levels and differential urinary microbiota abundance profiles between bladder cancer patients and healthy volunteers. All important and statistically significant information in the results chapter was also presented in very explanatory figures and tables making it more accessible to the readers.

Discussions were comprehensive, and presented research relevant to the subject. Current research on the subject was presented and results were compared to the ones evidenced by the authors, but also presented important first-time discoveries in their research - the detection of Rhodanobacter and Cutibacterium in urine samples might serve as highly specific biomarkers for bladder cancer.

The conclusions were clear and concise, and the conclusion was that bacteria such as Rhodanobacter, Cutibacterium, Alloscardovia, Moryella, and Anaeroglobus can be used as novel biomarkers for the diagnosis of bladder cancer, but more research is needed in order to identify the underlying mechanisms through which microbiota intervenes in carcinogenesis.

The methods chapter was clearly presented, and explained step by step the methodology used in order to get to the results. Research was sound, even though methods for microbiome research are not currently standardized.

In my opinion this study presents very innovative research, using DNA extraction, next-generation sequencing and bioinformatics analysis. It also brings new information useful that might prove useful in cancer research and, in the end, bring new diagnostic and therapy methods for clinicians.

For the reasons explained before i recommend for publishing after a very minor revision.  

Comments on the Quality of English Language

English language is very well used, there is a small mistake in page 2 rows 86-87.

Author Response

We would like to thank the reviewer for her/his positive comments and interest in our study. We are delighted that she/he found our research interesting and timely, as we would like to contribute to the growing body of knowledge on the role of the urinary microbiota in bladder cancer. The positive feedbacks regarding the clarity of our introduction, the significance of our findings, and the comprehensive presentation of our results are greatly appreciated. These comments are very encouraging to continuing our work on that way. As recommended, the missing word in the introduction has been added.